# Interspecific Competition and Vertical Niche Partitioning in Fiji's Forest Birds

**Alivereti N. Naikatini [1], Gunnar Keppel [2], Gilianne Brodie [1] and Sonia Kleindorfer [3,4,*]**

1 Institute of Applied Sciences, The University of the South Pacific, Laucala Campus, Private Mailbag, Suva 99999, Fiji; alivereti.naikatini@usp.ac.fj (A.N.N.); gilianne.brodie@usp.ac.fj (G.B.)
2 UniSA STEM and Future Industries Institute, University of South Australia, Mawson Lakes Campus, Adelaide, SA 5001, Australia; gunnar.keppel@unisa.edu.au
3 College of Science and Engineering, Flinders University, Adelaide, SA 5042, Australia
4 Konrad Lorenz Research Center for Behavior and Cognition, University of Vienna, 4645 Vienna, Austria
* Correspondence: sonia.kleindorfer@flinders.edu.au

**Abstract:** Charles Darwin proposed his 'principle of divergence' to account for changes in traits that could promote speciation and coexistence of diverse forms through occupation of different niches to reduce interspecific competition. We explore interspecific foraging behaviour overlap in Fiji's forest birds, and address two main questions: (1) Is there vertical stratification of foraging behavior? and (2) Is there evidence of interspecific competition driving the differences in foraging behaviour? We explore these questions across three foraging guilds, nectarivores (three species), insectivores (two species), and omnivores (two species), and find vertical portioning of foraging in each group. To investigate the effect of interspecific competition, we compared foraging heights of the Orange-breasted Myzomela (*Myzomela jugularis*) honeyeater on Viti Levu Island (where it coexists with two other honeyeater species) and Leleuvia Island (no other honeyeater species). On the main island Viti Levu, we found evidence for vertical niche partitioning within each foraging guild. On Leleuvia, with the 'one-species only foraging guild', Orange-breasted Myzomela occupied broader vertical foraging niche than on Viti Levu with two other competitor honeyeater species. This result supports the idea that vertical foraging height can be shaped by interspecific competition. The findings of this study support Darwin's principle of divergence in Fiji's forest birds for every foraging guild measured and adds to our understanding of the significance of interspecific competition and niche divergence for patterns of ecological speciation on islands.

**Keywords:** vertical stratification; resource niche partitioning; foraging height; foraging substrate; foraging technique

## 1. Introduction

In the 1960s, species composition on islands was famously modelled as the consequence of immigration, extinction, and habitat area [1,2], with faster immigration rates to islands closer to mainland source populations and faster extinction rates on smaller islands [3]. During the 2000s, the framing of island biogeography was expanded into the General Dynamical Model (GDM), whereby island age and environmental heterogeneity are introduced as additional factors that shape the ecology and biota of oceanic islands [3]. These ideas have received empirical support, for example comparison of vascular plants across 135 islands found an effect of environmental heterogeneity on species richness [4]. Islands harbour a higher proportion of endemic species compared to continents, and on Pacific islands, avian endemism and diversity is generally associated with the age, size, topography, and isolation of the island [5–8].

According to niche theory, species coexist when they are functionally adapted to different niches [9,10] and the diversity of niches has been shown to be positively related to species diversity, including different patterns of consumer strategy [11]. Different consumer

strategies include harvesting behaviour whereby species in a community or habitat may harvest the same resources but co-exist by adopting different foraging approaches, such as resource partitioning in space or time [11,12]. Resource niche partitioning can be achieved when animals forage on different substrates or use different techniques, and this has been studied across taxa, but especially in birds, on continental and island landscapes (e.g., [3,13]). In Jamaica, five species of migrant warblers overlap in diet but occupy different micro-habitats (branch tips, air space and bark) and use different foraging behaviours: one species used mainly gleaning, three species used both gleaning and hover gleaning, and one species used a combination of five foraging manoeuvres (sally strike, hover glean, sally, glean and flutter chase) [13]. Body size of the birds may be another factor that influences how and where animals forage; in a steppe-agrarian landscape in Spain, four species of birds (the Eurasian stone-curlew, *Burhinus oedicnemus* (Burhinidae), Red-legged partridge, *Alectoris rufa* (Phasianidae), Little bustard, *Tetrax tetrax* (Otididae), and Great bustard, *Otis tarda* (Otididae)), spanning medium (338 g) to large (18,000 g) size, occur in sympatry and select different micro-habitats (fallows used by Little bustards, abandoned lands by Eurasian stone-curlew, cereals by Great bustards, and field margins by Red-legged partridge) [14]. Other examples of sympatric occurrence of species assemblages with different foraging substrate and technique are known, such as three montane ground dwelling pheasants (Phasianidae) in the Qinghai-Tibet Plateau [15] or nine sympatric babbler species (Timaliidae) and three flycatcher-like species (Monarchidae) in the tropical rainforest of Malaysia [16,17]. Darwin's finches of the Galapagos Islands are an assemblage of 17 species that evolved over the past 1.5 million years with a range of beak sizes and shapes suited for different forms of resource extraction and foraging techniques [18]. Within the *Geospiza* ground finches, the different species are characterised by different beak sizes to crush seeds of different sizes and hardness; in the *Camarhynchus* tree finches, the different species are characterised by different beak curvature to extract insect prey from beneath bark or glean form the surface [19–21]. This divergent resource use in closely related avian lineages is a powerful lens through which to explore factors that shape sympatry on the one hand and fundamental factors that promote biodiversity on the other.

The distribution of species and partitioning of niches may be assessed by horizontal and vertical habitat use in the physical environment. The vertical stratification becomes more evident as species coexist by occupying specific vegetation strata associated with height, which reduces competition [22]. This has been documented not only for forest birds but for other taxa as well, such as forest arthropods that forage in five different vegetation layers (soil layer, litter layer, lower, canopy and upper canopy) in a tropical forest system [23], 51 species of bats that forage in four different vegetation layers (understory, canopy, above canopy including gaps, and opportunists which forage in both the lower and canopy strata) in a dry forest community in Central America [24], and 20 drosophilid species that showed vertically structured patterns of distribution in secondary and primary forests in Japan [25]. In the tropical dry forest of Peru, 29 of 37 bird species recorded foraged exclusively in particular forest layers and eight species foraged in four of the five recorded strata (emergent-upper canopy, canopy, subcanopy, shrub and ground) [26]. In the tropical forests of the larger islands of Melanesia, [27] there are three main vegetation layers where birds forage: the upper canopy (>30 ft) where most of the birds were recorded, the mid-canopy (10–30 ft), and the ground layer (<5 ft). In the larger mountain ranges of New Guinea, two whistlers coexist by using different layers of the forest, *Pachycephala soror* (understory) and *P. hyperythra* (>10 ft, canopy layers); in the North Coastal Ranges of New Guinea, two warbler species coexist, with *Sercornis virgatus* (understory) and *S. arfakianus* (middle canopy); and in Espiritu Santo island, Vanuatu, two species of white eyes are able to coexist in the forest edge habitats where *Zosterops lateralis* forages below 15 ft and *Z. flavifrons* forages above 15 ft [27]. In addition to resource availability and abiotic factors, forest physiogamy, tree architecture and vertical complexity of foliage structure also contribute to vertical stratification of animal communities in a forested habitat [23,25].

There are 66 land bird species in Fiji and 34 are endemic [28,29]. While we have a fair knowledge of the status of the native land birds of Fiji, basic knowledge of their ecology and behaviour is still lacking and mostly based on observations by early ornithologists. The two land bird species that have undergone some detailed ecological studies are the Red-vented Bulbul, *Pycnonotus cafer* [30] and the Red Avadavat, *Amandava amandava* [31], unfortunately both have been introduced to Fiji. A recent behavioural study carried out on a native species was by England [32] on the nesting behaviour of the endemic Natewa Silktail, *Lamprolia klinesmithi*. All other notes and observations recorded by earlier collectors and ornithologists are summarised by Watling [29] and the only reported work on the stratification of Fijian passerine birds was carried out by Langham [33]. Langham observed five vertical feeding zones in a Fijian rainforest: ground, undergrowth, lower storey, middle layer and canopy layer. He reported that five forest bird species mainly foraged in the three lower zones (*Turdus poliocephalus*, *Clytorhynchus vitiensis*, *Myiagra castaneigularis*, *Rhipidura layardi* and *Myzomela jugularis)* and six species mainly foraged in both the middle and upper canopy zones (*Zosterops lateralis*, *Zosterops explorator*, *Gymnomyza brunneirostris*, *Foulehaio procerior*, *Aplonis tabuensis* and *Clytorhynchus nigrogularis* [33].

In this study we examined the foraging behavior of forest birds in Fiji. We addressed two main questions: Firstly, is there vertical stratification of foraging behavior and secondly, is there evidence of interspecific competition driving the differences in foraging behaviour? We tested for differences in foraging behaviour for three species of nectarivores: Orange-breasted Myzomela (*Myzomela jugularis*), Kikau Honeyeater (*Foulehaio procerior*), Giant Honeyeater (*Gymnomyza brunneirostris*); two species of insectivores Slaty Monarch (*Mayrornis lesson*), Vanikoro Broadbill (*Myiagra vanikorensis*) and two species of omnivores; Fiji White-eye (*Zosterops explorator*), Silvereye (*Zosterops lateralis)*. All species co-occur in the same forest habitats on Viti Levu. We predicted that we will find evidence for different foraging behaviours (foraging height, foraging substrate, foraging technique) between species within each foraging guild. We tested for competitive release, by comparing foraging behaviour in one nectarivore species (Orange-breasted Myzomela) on two islands, Viti Levu Island that contains two other interspecific competitors (Kikau Honeyeater and Giant Honeyeater) and Leleuvia Island that lacks the other two nectarivore species. We predicted that there will be evidence for competitive release (broader vertical niche breadth use) in the island without other honeyeaters, as the vertical niche breadth should be constrained by the number of heterospecific species.

## 2. Materials and Methods

### 2.1. Study Sites

The foraging observation data were collected on Viti Levu Island from three forested habitats in two sites: a lowland mahogany forest located in the Colo-i-Suva Forest Park, a mid-elevation secondary forest, and a high-elevation primary forest, both located in the Mt. Koroyanitu National Heritage Park (Figure 1). We selected these sites as they are under some protection (ideal for long-term studies) and had forest cover that were representative of the forest types present on Viti Levu Island. The third site was Leleuvia Island, part of a "marine" reserve established in 2013 by the Leleuvia Island Resort management, where we only collected foraging data from the coastal forest on the one species of nectarivore present on the island, Orange-breasted Myzomela. This site being an island separated from Viti Levu by ocean, with more than 50% of the island being covered by natural coastal forest and under some conservation or protection status was ideal for this study.

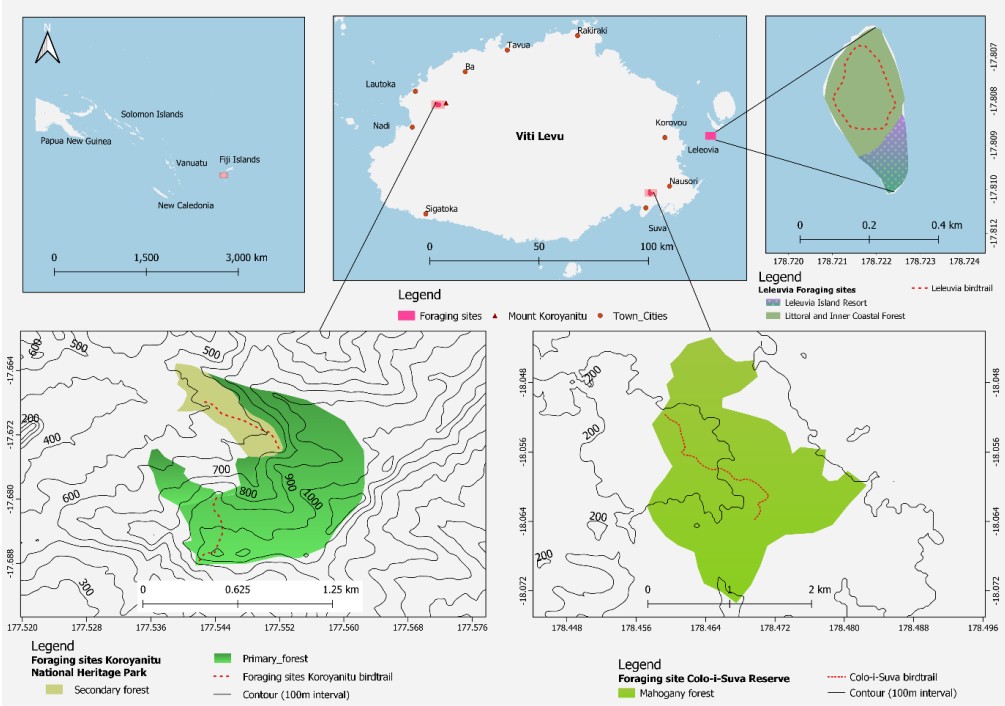

**Figure 1.** Survey area map, showing the location of the two main study sites; the Koroyanitu National Heritage Park located near the Nadi International Airport (western Viti Levu) and the Colo-i-Suva Forest Park located near Suva City (eastern Viti Levu). The Colo-i-Suva Forest Park is composed of lowland Mahogany plantation forest. The Koroyanitu National Heritage Park comprises two forest types, the mid-elevation secondary forest and high-elevation primary forest. The third site is a small coral islet, Leleuvia Island located to the east of Viti Levu Island. A good portion of the islet is still covered with littoral and inner coastal forest.

**Mt. Koroyanitu National Heritage Park.** Two study sites are located within the general area of this park, the high-elevation primary forest (−17°40′55.9″ S, 177°32′30.4″ E) and mid-elevation secondary forest (−17°40′86.0″ S, 177°32′38.2″ E). The Mt. Koroyanitu National Heritage Park covers an area of 25,000 ha with 700 documented plant species, 11 of which are endemic to the park [34]. The park was set up in the 1990s by the Ministry of Forests, Fiji Pine Limited, and iTaukei Land Trust Board (iTLTB) with the main objectives of the park being a conservation area and an eco-tourism area to generate income for the landowners [34–36]. It is one of the biggest reserves or protected areas in Fiji that uses a community-based management approach system [35,36].

The high-elevation forest at ≥900 m.a.s.l. is located along the track to Mt. Batilamu (Figure 1). The vegetation type is Montane Rain Forest which is still in its pristine state with *Agathis macrophylla* (Seem.) Silba (Araucariaceae) being the dominant tree species [37]. The mid-elevation forest at 500–800 m.a.s.l., located along the Savuione waterfall track (Figure 1), is a secondary forest, that has been altered by shifting agriculture, and landslides, mostly >50 years ago. *Pterocymbium oceanicum* A.C.Sm. (Sterculiaceae), *Bischofia javanica* Blume (Phyllanthaceae), and *Dendrocnide harveyi* (Seem.) Chew (Urticaceae) are the dominant tree species [38].

**Colo-i-Suva Forest Park.** The park is a low-elevation Mahogany plantation forest (−18°3′16.8″ S, 178°27′41.1″ E) that was established in 1963 (Figure 1). It covers an area of 92 ha and is part of the Colo-i-Suva Forest Reserve [35,39]. The Park was initially established as a Mahogany Plantation. However, the planted Mahogany (*Swietenia macrophylla* Ledoux & Lobato) was never logged, and the Park is now a conservation area and an eco-tourism site [36,40]. The introduced Mahogany is the dominant tree species, but extensive regeneration of native plant species has been documented as well [39–41].

**Leleuvia Island.** Leleuvia Island is a small coral islet (−17°48′31.9″ S, 178°34′18.8″ E), with an area of about 10 ha, located 13 km east of the nearest point on Viti Levu Island and is part of the Lomaiviti island group [42]. The dominant trees in the coastal littoral forest are beach mahogany (*Calophyllum inophyllum* L.), beach almond (*Terminalia littoralis* Pancher ex Guillaumin), coastal fig (*Ficus tinctoria* G. Forst) and other common species are beach trumpet (*Cordia subcordata* Lam), lantern tree (*Hernandia nymphaeifolia* (C. Presl) Fosberg), fish-poison tree (*Barringtonia asiatica* Druce), coconut palms (*Cocos nucifera* L.), beach gardenia (*Guettarda speciosa* L.) and pandanus (*Pandanus tectorius* Parkinson) [42,43]. The island has also been set up to cater for scientists and students both local and overseas interested in marine and terrestrial research and is considered "a living laboratory for environmental and marine education" [42].

### 2.2. Foraging Behaviour Data Collection

We standardised the foraging behaviour observation month to July because bird foraging activities in Fijian forest habitats are more readily observed at the onset of the breeding season, which peaks from June to September [29,44]. The transects in each site were about 500 m long and the foraging observations were carried out in the morning between 7 a.m. and 10 a.m. We recorded one observation per bird to avoid pseudo-replication of individual foraging behaviour. For each bird encountered along the transect, we noted its foraging substrate as (1) ground-ground, debris and litter, (2) foliage—live and dead foliage, (3) bark—live and dead bark, branches, (4) flower—live or dead flower, flower bud, (5) fruit—berries and, (6) moss—moss and lichen. The foraging technique was noted as (1) glean—remove prey from foliage surface, (2) pick—remove prey from non-foliage surface, (3) bite—ingest part of fooditem from surface, (4) probe—insert beack into substrate, (5) pry—use beak to lift substrate, and (6) chip-off—downward thrust of beak, usually repeating several times. The foraging substrates and techniques terminology and definitions were based on Schlotfeldt and Kleindorfer [45] and Myers [46]. In addition, the foraging height (meters above ground) was also recorded. Foraging height estimation was practiced using a laser pointer (LTI laser rangefinder) prior to field work using clearly visible trees on-campus at the University of the South Pacific. The laser rangefinder was first pointed at the base of the tree and then the top to compute two vertical angles, from which tree height was calculated. We calibrated among team members at the start of the field season and visually estimated tree height as meters above ground during field work. We used the first foraging observation as the first observation can be considered an independent observation [45,46].

### 2.3. Statistics and Data Analysis

In our first model we used the three foraging behaviours (height, substrate and technique) as our dependent variables and the seven targeted bird species from the three foraging guilds as our independent variables. We first tested for any significant difference between the two study sites on Viti Levu Island where we used foraging height of the targeted species as the dependent variable and the two sites as the independent variables. If there was no significat difference then we combined the data for two sites into one dataset and tested for potential differences in foraging height, foraging substrate and foraging techniques between the three nectarivore, two insectivore and two omnivore bird species. In our statictical analysis we used the General Linear Model (GLM) one way analysis of variance (ANOVA) in the IBM SPSS Statistics Software, v25 (Armonk, NY, USA) [47].

In our second model we used relative foraging height of the Orange-breasted Myzomela as our dependent variable and the two sites (mainland Viti Levu and Leleuvia island) as the independent variables. Since the average tree height on the mainland (Abaca and Colo-i-Suva 26 m) is taller than that on Leleuvia island (15 m) based on previous studies [37,48], we used the relative foraging height in the statistical analyses and graphical illustration. The relative foraging height was calculated by dividing the observed foraging height by the average tree height per site. To test for potential differences in the relative

foraging heights of the Orange-breasted Myzomela in the two sites we also used the the General Linear Model (GLM) one way analysis of variance (ANOVA) in the IBM SPSS Statistics Software, v25 [47].

## 3. Results

We collected 361 first foraging observations (see Table 1) of birds along the 500 m transect per study site on Viti Levu Island (referred to as 'mainland') over two years, and 40 observations of Orange-breasted Myzomela on Leleuvia island. Data were collected during July 2018 and 2019. We analysed 155 first foraging observations from three species of nectarivore, 57 from the two species of insectivore and 149 from the two species of omnivore across the study sites (Table 1). We recorded most of the foraging observations from Mt. Koroyanitu National Heritage Park (*n* = 295) compared to the Colo-i-Suva Forest Park (*n* = 66).

**Table 1.** The number of first foraging observations per species and survey site: Colo-i-Suva Forest Park, Koroyanitu National Heritage Park, and Leleuvia Island.

| Species | Colo-i-Suva (100 m.a.s.l.) | Koroyanitu (500–1000 m.a.s.l.) | Leleuvia (0 m.a.s.l.) |
|---|---|---|---|
| Nectarivore | | | |
| Orange-breasted Myzomela (*Myzomela jugularis*) | 26 | 22 | 40 |
| Kikau Honeyeater (*Foulehaio procerior*) | 9 | 85 | |
| Giant Honeyeater (*Gymnomyza brunneirostris*) | 2 | 11 | |
| Insectivore | | | |
| Slaty Monarch (*Mayrornis lesson*) | 4 | 17 | |
| Vanikoro Broadbill (*Myiagra vanikorensis*) | 10 | 26 | |
| Omnivore | | | |
| Fiji White-eye (*Zosterops explorator*) | 15 | 74 | |
| Silvereye (*Zosterops lateralis*) | 0 | 60 | |

### 3.1. Foraging Location and Species Differences

On the mainland, there was a significant difference in foraging height across species, but no significant effect of foraging location (lowland or mid-elevation forest) (General Linear Model: location: F = 2.093, df = 1,423, *p* = 0.149; species: F = 16.306, df = 6,423, *p* < 0.001; interaction term location × species; F = 0.904, *p* = 0.492). Therefore, we combined foraging data observations for both locations and examined species differences within the three foraging guilds.

### 3.2. Comparisons within Foraging Guilds

Mainland nectarivore species differed significantly in foraging height across the three species (General linear model, F = 32.216, df = 1,193, *p* < 0.001). Tukey Post-hoc tests showed that all pairwise comparisons were statistically significant (*p* < 0.001). Foraging height was lowest in the Orange-breasted Myzomela, highest in the Giant Forest Honeyeater, and intermediate in Kikau Honeyeater (Figure 2). For substrate category, there was a significant difference across the three species (Likelihood ratio, df = 6, *p* = 0.023; Table 2). Flowers were the primary substrate used by all three species, Giant Honeyeater (50%), Orange-breasted

Honeyeater (49.1%) and Kikau Honeyeater (48.7%). The Giant Honeyeater used bark as the secondary substrate (44.4%), whereas leaves and bark were secondary substrates for the Orange-breasted Myzomela (26.3% and 24.3%) and Kikau Honeyeater (20.2% and 23.5%). The three nectarivores never foraged on the ground. However, the Kikau Honeyeater foraged on four substrates (flower, bark, leaves, air) and the other two nectarivores foraged only on three substrates (flower, bark and leaves). For foraging technique, there was a significant difference across the three species (Likelihood ratio, df = 8, *p* < 0.001). Probing (primary) and gleaning (secondary) were the main foraging techniques used by all the three species. In addition, the Giant Honeyeater also used biting, the Kikau Honeyeater used chip-off pry and sally hover glean, and the Orange-breasted Myzomela used biting and sally hover glean (Table 3).

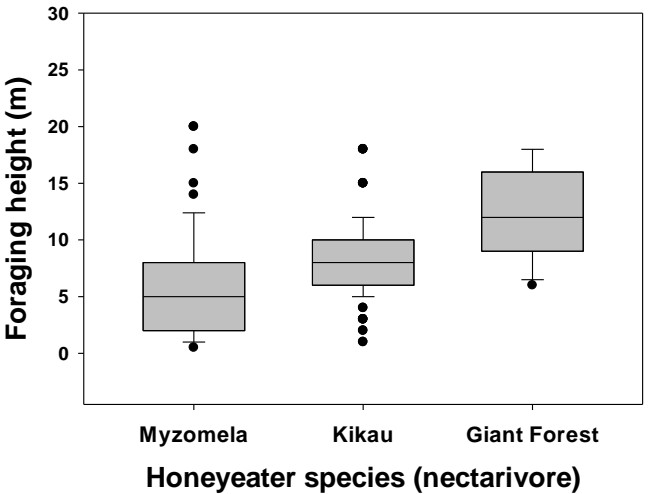

**Figure 2.** Box plots (median score, interquartile range, whiskers) for foraging height (m) in three sympatric honeyeater species on Viti Levu, Fiji. The species are Orange-breasted Myzomela (*Myzomela jugularis*) (*n*= 48), Kikau Honeyeater (*Foulehaio procerior*) (*n* = 94), and Giant Forest Honeyeater (*Gymnomyza brunneirostris*) (*n* = 13).

**Table 2.** Foraging substrate (percentage of observations) used by birds from seven species in three foraging guilds (see Table 1 for sample sizes). For Myzomela, we show data for the mainland site (with competitor honeyeater species) and the island site (without competitor honeyeater species). Foraging substrate in Myzomela did not differ significantly across the two sites (see results).

| Substrate | Nectarivore | | | | Insectivore | | Omnivore | |
|---|---|---|---|---|---|---|---|---|
| | **Myzomela Island** | **Myzomela Mainland** | **Kikau** | **Giant** | **Slaty** | **Vanikoro** | **White-Eye** | **Silver-Eye** |
| Ground | 0 | 0 | 0 | 0 | 0 | 14% | 0 | 0 |
| Bark | 53% | 25% | 24% | 44% | 57% | 32% | 39% | 38% |
| Leaves | 2.5% | 21% | 20% | 6% | 39% | 27% | 45% | 56% |
| Flower | 38% | 54% | 49% | 50% | 4% | 16% | 17% | 7% |
| Air | 8% | 0 | 8% | 0 | 0 | 11% | 0 | 0 |

Mainland insectivore species differed significantly in foraging height (General Linear Model: F = 38.195, df = 1,66, *p* < 0.001). Foraging height was lower in Vanikoro than in Slaty Monarch (Figure 3). Foraging substrate differed significantly between the two species (Likelihood Ratio df = 4, *p* = 0.003). Whereas both species mostly foraged on bark (Slaty Monarch 57.1%, Vanikoro 32.4%), Vanikoro also regularly foraged on four other substrates (ground 13.5%, leaves 27%, flower 16.2%, air 10.8%) (Table 2). Foraging technique also differed significantly between the two insectivore species (Likelihood ratio df = 4, *p* < 0.001).

While the dominant foraging technique in both species was gleaning, Vanikoro was also observed to bite, probe, pry, sally, and hover glean (Table 3).

**Table 3.** Foraging technique (percentage of observations) used by birds from seven species in three foraging guilds (see Table 1 for sample sizes). In the case of Myzomela, we show data for the mainland site (with competitor honeyeater species) and the island site (without competitor honeyeater species). Foraging technique in Myzomela did not differ significantly across the two sites (see results).

| Technique | Nectarivore | | | | Insectivore | | Omnivore | |
|---|---|---|---|---|---|---|---|---|
| | Myzomela Island | Myzomela Mainland | Kikau | Giant | Slaty | Vanikoro | White-Eye | Silver-Eye |
| Glean | 55% | 40% | 30% | 17% | 93% | 31% | 81% | 90% |
| Bite | 15% | 13% | 0 | 6% | 3% | 18% | 8% | 2% |
| Probe | 20% | 44% | 55% | 78% | 4% | 15% | 11% | 8% |
| Chip off + Pry | 0 | 0 | 5% | 0 | 0 | 3% | 0 | 0 |
| Sally + Hoverglean | 10% | 4% | 10% | 0 | 0 | 33% | 0 | 0 |

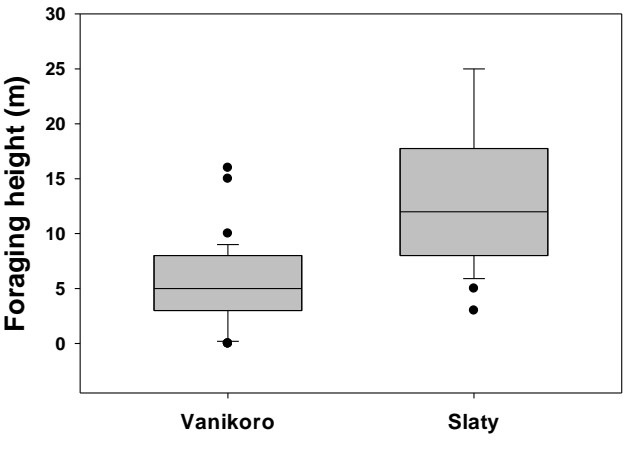

**Figure 3.** Box plots (median score, interquartile range, whiskers) for foraging height (m) in two sympatric flycatcher species on Viti Levu, Fiji. The species are Vanikoro Broadbill (*Myiagra vanikorensis*) (*n* = 36) and Slaty Monarch (*Mayrornis lessoni*) (*n* = 21).

Mainland omnivore species differed significantly in foraging height (General Linear Model: F = 55.373, df = 1,162, *p* < 0.001). Foraging height was lower in White-eye and higher in Silvereye (Figure 4). Foraging substrate was comparable in the two species (Likelihood ratio df = 2, *p* = 0.134) (Table 2) and both species were observed to forage on bark, leaves and flowers. Foraging technique was also comparable in the two species (Likelihood ratio df = 2, *p* = 0.109), and both species mostly used gleaning as the main technique (White-eye 81.3%, Silvereye 90.3%) with some biting and probing (Table 3).

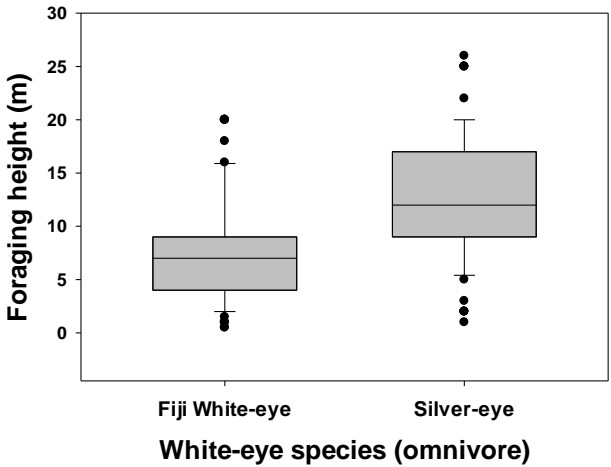

**Figure 4.** Box plots (median score, interquartile range, whiskers) for foraging height (m) in two sympatric *Zosterops* species on Viti Levu, Fiji. The species are Fiji White-eye (*Zosterops explorator*) (*n* = 89) and Silvereye (*Zosterops lateralis*) (*n* = 60).

*3.3. Myzomela Foraging in Areas with and without Other Honeyeater Species*

We compared foraging height and behaviour in Myzomela in mainland forest with three sympatric honeyeater species (Myzomela, Kikau, Giant Honeyeater) and in coastal forest where Myzomela was the only honeyeater. In the coastal forest without competitor species, Myzomela had higher foraging height than in mainland forest with competitor honeyeater species (General linear model F = 9.50, df = 1,85, *p* = 0.003) (Figure 5). But foraging substrate and technique did not differ significantly across sites with and without competitors (substrate: F = 2.23, *p* = 0.139; technique: F = 0.711, *p* = 0.402).

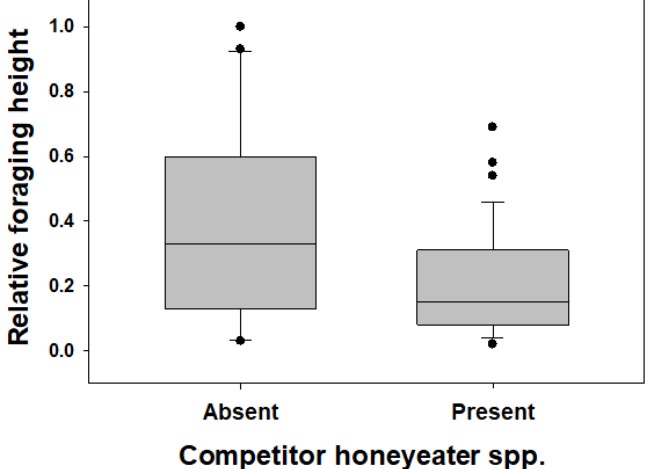

**Figure 5.** Box plots (median score, interquartile range, whiskers) showing foraging height relative to average tree height in Orange-breasted Myzomela in the site with only Myzomela (Leleuvia Island; competitor honeyeaters absent) and in the sites with Myzomela, Kikau and Giant honeyeater (Viti Levu mainland; competitor honeyeaters present). The average forest tree height is 15 m on Leleuvia Island and 26 m on Viti Levu mainland; data are shown as observed foraging height/average tree height. The statistical analysis was done on LN of observed foraging height, and both average foraging height and relative foraging height were significantly higher in the site without honeyeater competitors.

## 4. Discussion

We found evidence that, within the three guilds, species differed in average foraging height. In addition, several within-guild species also differed in foraging substrate and technique. Thus, we conclude that there is evidence for vertical stratification of foraging

behaviour of different species within the same guild with further niche partitioning from differences in foraging substrate and technique. In the second study, the relative foraging height was significantly higher in the study site that only had *Myzomela* and lacked competitor honeyeater species—notably species that occupy higher foraging heights where they occur. The fact that *Myzomela* foraging substrate and technique were similar in areas with and without nectarivore competitors, but that foraging height increased without competitors that usually occupy higher foraging sites, points to competitor release for vertical niche breadth. The findings support the idea that vertical foraging height can be shaped by interspecific competition, as has been found in other studies [22].

Interspecific competition can be costly and individuals that avoid competitors, for example by using different niches, can, over longer periods, come to inhabit new niches and/or become niche specialists. Vertical stratification of foraging substrate is one mechanism that can favour coexistence of different species that forage at different forest heights or layers [22]. Vertical foraging stratification has been observed in arthropods [23], bats [24] and birds [25,26]. Although Viti Levu Island is considered young geologically and was exposed terrestrially about 26 million years ago [49,50], it is much older than the Galapagos Islands (ca 3 million years, with Darwin's finch evolution ~1.5 million years later) [18]. Irrespective of the age of the habitat, adaptive radiations can be fast, such as the rapid ecological, morphological, and foraging behaviour niche diversification that generated 17 Darwin's finch species in about 1.5 million years [18].

In this study, we find consistent evidence for species using different vertical niches within guilds, which suggests that vertical partitioning could be favoured by reduced interspecific competition. The observations also tend to agree with the observations recorded by Langham [33] for the honeyeaters where he observed Orang-breasted *Myzomela* in the three lower strata (ground, undergrowth and lower storey) and Kikau Honeyeater and Giant Honeyeater in the middle and upper canopy strata. The three nectarivores in this study are all endemic to Fiji and are the main nectarivore species present on Viti Levi island. The Orange-breasted *Myzomela* is most widespread (occurs throughout Fijian archipelago), the Kikau is restricted to Viti Levu and parts of Vanua Levu, and the Giant Honeyeater is endemic to Viti Levu Island alone. Only the Giant Honeyeater might be considered a true forest species as it is only present in intact primary and mature secondary forests, while the other two honeyeater species occur in almost all terrestrial habitats present on Viti Levu [29,51]. When all honeyeaters are present in a forest habitat, they exhibit vertical foraging or resource partitioning with Giant Honeyeater feeding in the canopy, Kikau in the mid-canopy, and *Myzomela* in the lower canopy (Figure 2). Forests are thought to have dominated the Fiji archipelago for millions of years and therefore it is possible that the three nectarivore species have a long history of sympatric association in forests [52].

A total of 16 insectivorous bird species are recorded on Viti Levu, comprising 28% of the total birds species present on the island but in this study our focus was only on two species. The Vanikoro Flycatcher is native to the Solomons and Fiji and occurs in almost all the terrestrial habitats present on Viti Levu. The second insetivore species was the Slaty Monarch, which is endemic to Fiji and occurs in intact forest and mature secondary forest [29]. As with the island endemic Giant Honeyeater, the endemic Slaty Monarch forages mostly in the canopy and the native Vanikoro in the mid-canopy (Figure 3). There are five omnivorous bird species recorded for Viti Levu, however in this study our focus was on the two native species. The native Silvereye mostly occurs in secondary or disturbed forest habitats compared to the endemic Fiji White-eye that is mostly associated with intact or mature secondary systems. However, when they are recorded together, mostly along the forest edges, the native Silvereye mostly forages in the canopy and the endemic White-eye in the mid-canopy (Figure 4). This is different in Espiritu Santo Island, Vanuatu, where the native Silvereye forages in the sub-canopy and the endemic Vanuatu White-eye (*Zosterops flavifrons*) in the canopy [26].

Therefore, for all guilds investigated, there is a pattern that endemic species tend to forage in the canopy and native species within guilds tend to forage in the lower and

mid-canopy. The endemic Fiji White-eye appears to be an exception to this trend. However, our observations could be biased because we worked mostly in secondary forest and the Fiji White-eye could occur mostly in the canopy of primary forest. The omnivores being generalist feeders or generalist guild might also exhibit less competition when compared to the specialised guilds like the insectivores, and nectarivores. Further investigation and testing with larger sample sizes is required to address this. Our observations for the insectivorous birds could also be biased as we only focussed on two species instead of all the 16 insectivore species present on Viti Levu. Further investigation needs to address the foraging behaviour of insectivorous birds in Fijis forest habitats looking at more insectivore species instead of just two species.

Several endemic species in this study (nectivorous Giant Honeyeater, insectivorous Slaty Monarch, omnivorous Fiji White-eye) exhibited characteristics that suggest they are forest specialists because they mostly occur in intact and mature secondary forest. In contrast, the other species (nectivorous Orange-breasted Myzomela, nectivorous Kikau Honeyeater, insectivorous Vanikoro Flycatcher, omnivorous Silvereye) have characteristics of generalist species as they occur in forested and non-forested areas on Viti Levu Island. These generalist species would have higher chances of surviving on Viti Levu Island as they have been able to adapt to almost all the terrestrial habitats present on the island compared to the forest specialists. *Myzomela* honeyeaters in New Guinea and New Britain are restricted to forest habitats and face competition from same-sized honeyeaters in the genus *Nectarina*, however in Fiji where *Myzomela* is the only small honeyeater, it forages in all habitats including open area and forested habitats [26].

Competitor or ecological release is expected to result in measurable differences in behaviour in the absence of competitors. Our study found support for this idea in Orange-breasted Myzomela on Leleuvia Island, where the species occupied higher average foraging height when the two competitor honeyeater species that usually occupy the highest foraging layers were missing. Competitor release has been observed and documented in other island systems such as the avifauna on Mona Island and Guanico Forest in Puerto Rico [53]. Acoustic space can also be freed from competitor release and studies have found differences in frequency bandwidth based on presence of competitor species in birds on Sao Tome Island and Mt Cameroon [54]. Island and mainland foraging breadth also differed according to predictions of competitor release in honeyeaters studied in South Australia [46]. It would be interesting to test if body size correlates with vertical forest partitioning. If larger birds within each guild tend to occupy the canopy, and smaller birds the understory, this raises questions about processes that could favour particular body sizes in relation to resource distribution and/or the costs of resource defense in the different forest vegetation layers [55,56]. One aspect not considered in this study was diet, and future research could measure the inter-guild functional roles of each species, also in relation to diet and seed dispersal, for example [57].

**Author Contributions:** Conceptualization, S.K.; methodology, S.K. and G.K.; formal analysis, S.K., G.K. and A.N.N.; investigation, S.K. and A.N.N.; data curation, S.K.; writing—original draft preparation, A.N.N.; writing—review and editing, S.K., G.K., G.B. and A.N.N.; supervision, S.K., G.K. and G.B.; project administration, S.K.; funding acquisition, S.K. and G.K. All authors have read and agreed to the published version of the manuscript.

**Funding:** New Colombo Plan initiative through the Australia Government Department of Foreign Affairs and Trade, Royal Commonwealth Society of South Australia and University Research Council of the University of the South Pacific.

**Institutional Review Board Statement:** Not applicable.

**Informed Consent Statement:** Not applicable.

**Data Availability Statement:** Data are available upon request from the corresponding author.

**Acknowledgments:** We acknowledge and thank all the students from Flinders University and University of South Australia who participated in the surveys. We thank the volunteers from NatureFiji-

MareqetiViti and The University of the South Pacific who took part in the surveys. We are grateful to the Fiji Ministry of Forestry and Abaca Village community for field assistance and permission to access and conduct bird surveys in the Colo-i-Suva Mahogany Reserve and the Koroyanitu National Heritage Park. We also thank and acknowledge the management team, of Leleuvia Island Resort for allowing us to carry out bird surveys on Leleuvia Island. We would particularly like to acknowledge the late Colin Philp, who was the Resort Manager at Leleuvia Island, for his strong support during our stay and field work at Leleuvia Island.

**Conflicts of Interest:** The authors declare no conflict of interest. The funders had no role in the design of the study; in the collection, analyses, or interpretation of data; in the writing of the manuscript, or in the decision to publish the results.

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
