# Peer review of "Interspecific Competition and Vertical Niche Partitioning in Fiji’s Forest Birds"

_diversity, doi:10.3390/d14030223_

Round 1
Reviewer 1 Report
The manuscript investigates a classical ecological concept of foraging niche partitioning between a few Fiji´s forest bird species. What I really like on the paper is the focus on vertical locations of birds within the forest, which is still not adequately explored. By the way, I suggest putting “height” into the title. The authors provided simple but clear and valuable insights that deserve to be published. However, I think that some text reduction and adjustments must be done before.
- The Introduction starts from a very general perspective, which is not bad, but it is not needed, and it prolongs the text non-adequately. Thus, I would suggest shortening the Introduction by at least 30% so that it focuses more on the question tested and less on general stuff related. That can be mentioned just briefly.
- Page 2, line 56. I think that calling abiotic factors a “physiological niche” is oversimplification in the context of the paper. Habitat preferences are in fact the most important aspect of the niche in birds and those are not necessarily linked to physiological requirements and distribution of a habitat is determined by abiotic conditions such as temperature and rainfall.
- The similar reduction can be applied to details on Fiji birds on page 4, which is not needed for the paper.
- Number of species tested is relatively low thus in does not allow to understand the community context. However, I think that few sentences in the Discussion about how the communities “around” the focal species look like can help.
- The statistical analyses are simple but looks appropriate. But I miss detailed information about the models and for example interactions between variables. I see that the data set in not a big one, however it seems to be clear that for instance tactics may differ depending on the location along the vertical axis of the forest. Please, could you provide some more complete overview of the model results. And potentially discuss at least the interactions.
- I find the Table 2 redundant. It can be explained directly in the text and some terms even do not need further explanation (Ground=Ground, Fruit=Berries etc.).
- I suggest sending Tables 3 and 4 to the Supplementary material. The information in them is of course interesting but not directly related to main results.
- The foraging difference between omnivores deserves more discussion as one can expect that omnivores being not so specialized could in fact go around the competition more easily than specialists.
Overall, I like the topic of the paper but before I can recommend it for publication, I suggest reduction of the text and tables, maybe put it more into the short note style. Provide more details about statistical models and highlight the height above the ground in the line of the text as well as in the title as it is the main result.
Author Response
The manuscript investigates a classical ecological concept of foraging niche partitioning between a few Fiji´s forest bird species. What I really like on the paper is the focus on vertical locations of birds within the forest, which is still not adequately explored. By the way, I suggest putting “height” into the title.
Response: Thank you for your positive comments and helpful suggestions. We have inserted ‘vertical’ into the title, so the title now reads: “Interspecific competition and vertical niche partitioning in Fiji’s forest birds”.
The authors provided simple but clear and valuable insights that deserve to be published. However, I think that some text reduction and adjustments must be done before. Overall, I like the topic of the paper but before I can recommend it for publication, I suggest reduction of the text and tables, maybe put it more into the short note style. Provide more details about statistical models and highlight the height above the ground in the line of the text as well as in the title as it is the main result.
Our response to the other comments and helpful suggestions highlighted are addressed individually below.
The Introduction starts from a very general perspective, which is not bad, but it is not needed, and it prolongs the text non-adequately. Thus, I would suggest shortening the Introduction by at least 30% so that it focuses more on the question tested and less on general stuff related. That can be mentioned just briefly.
Response: Yes, we agree with the suggestion and have shortened the Introduction to keep the manuscript focused on the question.
Page 2. line 56. I think that calling abiotic factors a “physiological niche” is oversimplification in the context of the paper. Habitat preferences are in fact the most important aspect of the niche in birds and those are not necessarily linked to physiological requirements and distribution of a habitat is determined by abiotic conditions such as temperature and rainfall.
Response: This sentence has been deleted from the manuscript when we shortened the ‘Introduction’.
Page 4.The similar reduction can be applied to details on Fiji birds on page 4, which is not needed for the paper.
Response: This paragraph has also been shortened from 320 words to 200 words.
Page 5. Number of species tested is relatively low thus in does not allow to understand the community context. However, I think that few sentences in the Discussion about how the communities “around” the focal species look like can help.
Response: We have added new sentences in the Discussion section accordingly and provided the number of species from the same foraging guilds present in the study sites.
Page 9. The statistical analyses are simple but looks appropriate. But I miss detailed information about the models and for example interactions between variables. I see that the data set is not a big one, however it seems to be clear that for instance tactics may differ depending on the location along the vertical axis of the forest. Please, could you provide some more complete overview of the model results. And potentially discuss at least the interactions.
Response: We have added new sentences to provide detail of the model and variables used to make clear to the reader.
Page 8.I find the Table 2 redundant. It can be explained directly in the text and some terms even do not need further explanation (Ground=Ground, Fruit=Berries etc.).
Response: Agree, this table has been deleted as suggested.
Page 11.I suggest sending Tables 3 and 4 to the Supplementary material. The information in them is of course interesting but not directly related to main results.
Response: We would like to keep the two tables in the results sections, as they are of benefit to the reader and as small summary tables, or we do not believe they need to be placed as supplementary material.
Page 15. The foraging difference between omnivores deserves more discussion as one can expect that omnivores being not so specialized could in fact go around the competition more easily than specialists.
Response: Yes, we agree with this point of view and mentioned it on page 15 and 16 and highlighting that, further investigations are required in the future L 483-486.
Page 2.The Introduction starts from a very general perspective, which is not bad, but it is not needed, and it prolongs the text non-adequately. Thus, I would suggest shortening the Introduction by at least 30% so that it focuses more on the question tested and less on general stuff related. That can be mentioned just briefly.
Response: Yes, we agree with the suggestion and have shortened the Introduction to keep the manuscript focused on the question.
Page 2. line 56. I think that calling abiotic factors a “physiological niche” is oversimplification in the context of the paper. Habitat preferences are in fact the most important aspect of the niche in birds and those are not necessarily linked to physiological requirements and distribution of a habitat is determined by abiotic conditions such as temperature and rainfall.
Response: This sentence has been deleted from the manuscript when we shortened the ‘Introduction’.
Page 4.The similar reduction can be applied to details on Fiji birds on page 4, which is not needed for the paper.
Response: This paragraph has also been shortened from 320 words to 200 words.
Page 5. Number of species tested is relatively low thus in does not allow to understand the community context. However, I think that few sentences in the Discussion about how the communities “around” the focal species look like can help.
Response: We have added new sentences in the Discussion section accordingly and provided the number of species from the same foraging guilds present in the study sites.
Page 9. The statistical analyses are simple but looks appropriate. But I miss detailed information about the models and for example interactions between variables. I see that the data set is not a big one, however it seems to be clear that for instance tactics may differ depending on the location along the vertical axis of the forest. Please, could you provide some more complete overview of the model results. And potentially discuss at least the interactions.
Response: We have added in new sentences to provide detail of the model and variables used to make clear to reader.
Page 8.I find the Table 2 redundant. It can be explained directly in the text and some terms even do not need further explanation (Ground=Ground, Fruit=Berries etc.).
Response: Agree, this table has been deleted as suggested.
Page 11.I suggest sending Tables 3 and 4 to the Supplementary material. The information in them is of course interesting but not directly related to main results.
Response: We would like to keep the two tables in the results sections, as they are of benefit to the reader and as small summary tables, or we do not believe they need to be placed as supplementary material.
Page 15. The foraging difference between omnivores deserves more discussion as one can expect that omnivores being not so specialized could in fact go around the competition more easily than specialists.
Response: Yes, we agree with this point of view and mentioned it on page 15 and 16 and highlighting that, further investigations are required in the future L 483-486.
Reviewer 2 Report
The Manuscript is a nice study on poorly known bird species in island ecosystems.
-
Although I am not a native speaker, I found some problems with English flowing and I suggest language polishing.
-
Introduction is too long and partly unfocused. Please, keep clear aims and predictions and delete all the other digressions, which make the reader lost.
-
Lines 52-53. Please, add a reference.
-
Line 56. Too much spaces after the bracket.
-
Lines 72-76. Add scientific names, please.
-
Lines 149-166. Please improve this part to put your paper into a more hypothesis-driven context. Why is your work important? Be clear with aims and show also your predictions in line of previous literature.
-
Lines 168-175. This is a result, together with the table. Start with study area and only describe (in the methods) how you collected data.
-
Table 1 and Study site. Add scientific names.
-
Scientific names of plants requires the name of the descriptor.
-
Lines 244 and following. Please, be clear here: how long was each transect? How long did you stay watching birds per survey? How many surveys per month?
-
Are you sure you covered a representative part of the study site to get reliable results? Please clarify.
-
Lines 369-371. This is an aim, not discussion.
-
I found some scientific names in the Discussion not italicized. Please, check and correct.
Author Response
The Manuscript is a nice study on poorly known bird species in island ecosystems.
Response: Thank you.
1.Although I am not a native speaker, I found some problems with English flowing and I suggest language polishing.
Response: Spell and grammar check completed
2. Introduction is too long and partly unfocused. Please, keep clear aims and predictions and delete all the other digressions, which make the reader lost.
Response: The introduction section has been shortened from 1574 words to 1260 words.
3. Lines 52-53. Please, add a reference.
Response: This sentence has been deleted.
4. Line 56. Too many spaces after the bracket.
Response: This has been edited out.
5. Lines 72-76. Add scientific names, please.
Response: Scientific names have been added
6. Lines 149-166. Please improve this part to put your paper into a more hypothesis-driven context. Why is your work important? Be clear with aims and show also your predictions in line of previous literature.
Response: Completed. We have improved this section of the manuscript as per the reviewers’ instructions.
7. Lines 168-175. This is a result, together with the table. Start with study area and only describe (in the methods) how you collected data.
Response: Table 1 and the associated paragraph have been moved to the results section.
8. Table 1 and Study site. Add scientific names.
Response: Scientific names have been added.
9. Scientific names of plants requires the name of the descriptor.
Response: Descriptor names have been added.
10. Lines 244 and following. Please, be clear here: how long was each transect? How long did you stay watching birds per survey? How many surveys per month?
Response: Done
11. Are you sure you covered a representative part of the study site to get reliable results? Please clarify.
Response: Yes, we selected forest habitats that were representative of the study sites. This has already been clarified in the manuscript.
11. Lines 369-371. This is an aim, not a discussion.
Response: This section has been deleted as it is repeated.
12. I found some scientific names in the Discussion not italicized. Please, check and correct.
Response: Yes, in the discussion section we use the Myzomela as a common name and not the scientific name. In the discussion where Myzomela is mentioned alone, we have italicized it and where it appears in the common name: Orange-breasted Myzomela we have not italicized it.
Round 2
Reviewer 2 Report
Authors have now addressed all of my previous comments, and the MS can be accepted for publication on DIVERSITY.